# Explaining V1 Properties with a Biologically Constrained Deep Learning Architecture

**Galen Pogoncheff**
Department of Computer Science
University of California, Santa Barbara
Santa Barbara, CA 93106
galenpogoncheff@ucsb.edu

**Jacob Granley**
Department of Computer Science
University of California, Santa Barbara
Santa Barbara, CA 93106
jgranley@ucsb.edu

**Michael Beyeler**
Department of Computer Science
Department of Psychological & Brain Sciences
University of California, Santa Barbara
Santa Barbara, CA 93106
mbeyeler@ucsb.edu

## Abstract

Convolutional neural networks (CNNs) have recently emerged as promising models of the ventral visual stream, despite their lack of biological specificity. While current state-of-the-art models of the primary visual cortex (V1) have surfaced from training with adversarial examples and extensively augmented data, these models are still unable to explain key neural properties observed in V1 that arise from biological circuitry. To address this gap, we systematically incorporated neuroscience-derived architectural components into CNNs to identify a set of mechanisms and architectures that more comprehensively explain V1 activity. Upon enhancing task-driven CNNs with architectural components that simulate center-surround antagonism, local receptive fields, tuned normalization, and cortical magnification, we uncover models with latent representations that yield state-of-the-art explanation of V1 neural activity and tuning properties. Moreover, analyses of the learned parameters of these components and stimuli that maximally activate neurons of the evaluated networks provide support for their role in explaining neural properties of V1. Our results highlight an important advancement in the field of NeuroAI, as we systematically establish a set of architectural components that contribute to unprecedented explanation of V1. The neuroscience insights that could be gleaned from increasingly accurate in-silico models of the brain have the potential to greatly advance the fields of both neuroscience and artificial intelligence.

## 1 Introduction

Many influential deep learning architectures and mechanisms that are widely used today, such as convolutional neural networks (CNNs) [1] and mechanisms of attention [2–5], draw inspiration from biological intelligence. Despite decades of research into computational models of the visual system, our understanding of its complexities remains far from complete. Existing neuroscientific models of the visual system (e.g., generalized linear-nonlinear models [6–9]) are often founded upon empirical observations from relatively small datasets, and are therefore unlikely to capture the true complexity of the visual system. While these models have successfully explained many properties of neural response to simple stimuli, their simplicity does not generalize to complex image stimuli [10].

37th Conference on Neural Information Processing Systems (NeurIPS 2023).

Following their astounding success in computer vision, task-driven CNNs have recently been proposed as candidate models of the ventral stream in primate visual cortex [11–15], offering a path towards models that can explain hidden complexities of the visual system and generalize to complex visual stimuli. Through task-driven training alone (and in some cases, training a linear read-out layer [12–17]), representations that resemble neural activity at multiple levels of the visual hierarchy have been observed in these models [16]. With the emergence of such properties, CNNs are already being used to enhance our knowledge of processing in the ventral stream [18].

Despite these advancements, CNNs that achieve state-of-the-art brain alignment are still unable to explain many properties of the visual system. Most traditional CNNs omit many well known architectural and processing hallmarks of the primate ventral stream that are likely key to the development of artificial neural networks (ANNs) that help us decipher the neural code. The development of these mechanisms remains an open challenge. A comprehensive understanding of neural processing in the brain (for instance, in the ventral stream) could in turn contribute to significant leaps in artifical intelligence (AI) – an established goal of NeuroAI research [19, 20].

In this work, we take a systematic approach to analyzing the hallmarks of the primate ventral stream that improve model-brain similarity of CNNs. We formulate architectural components that simulate these processing hallmarks within CNNs and analyze the population-level and neuron-level response properties of these networks, as compared to empirical data recorded in primates. Specifically:

- We enrich the classic ResNet50 architecture with architectural components based on neuroscience foundations that simulate cortical magnification, center-surround antagonism, local filtering, and tuned divisive normalization and show that the resulting network achieves top V1-overall score on the integrative Brain-Score benchmark suite [16].

- Although some of these components have been studied before in isolation, here we demonstrate their synergistic nature through a series of ablation studies that reveal the importance of each component and the benefits of combining them into a single neuro-constrained CNN.

- We analyze the network parameters and stimuli that activate neurons to provide insights into how these architectural components contribute to explaining primary visual cortex (V1) activity in non-human primates.

## 2   Background and Related Work

**Model-Brain Alignment**   One central challenge in the field of NeuroAI is the development of computational models that can effectively explain the neural code. To achieve this goal, artificial neural networks must be capable of accurately predicting the behavior of individual neurons and neural populations in the brain. The primary visual cortex (V1) is one of the most well studies areas of the visual system, with modeling efforts dating back to at least 1962 [21]—yet many deep learning models still fall short in explaining its neural activity.

The Brain-Score integrative benchmark [16] has recently emerged as a valuable tool for assessing the capabilities of deep learning models to explain neural activity in the visual system. This suite of benchmarks integrates neural recording and behavioral data from a collection of previous studies and provides standardized metrics for evaluating model explainability of visual areas V1, V2, V4, and IT, as well as additional behavioral and engineering benchmarks.

Although CNNs draw high-level inspiration from neuroscience, current architectures (e.g., ResNet [22] and EfficientNet [23]) bear little resemblance to neural circuits in the visual system. While such differences may not necessarily hinder object recognition performance, these networks still fall short in mimicking many properties of highly capable visual systems. Although there may be many paths towards next-generation AI, foundational studies that have successfully merged foundations of neuroscience and AI have shown promising improvements to traditional ANNs [24–26].

**Center-Surround Antagonism**   As early as in the retina, lateral inhibitory connections establish a center-surround antagonism in the receptive field (RF) of many retinal cell types, which is preserved by neurons in the lateral geniculate nucleus and the visual cortex. In the primate visual system, this center-surround antagonism is thought to facilitate edge detection, figure-ground segregation, depth perception, and cue-invariant object perception [27–30], and is therefore a fundamental property of visual processing.

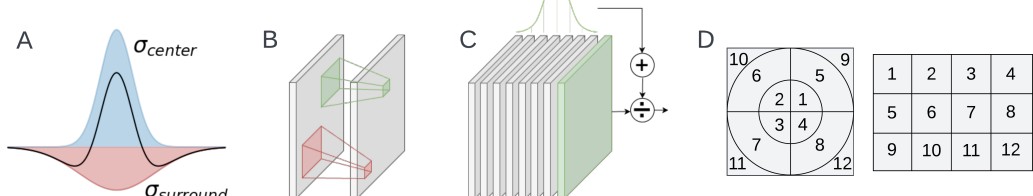

Figure 1: Design patterns of neuro-constrained architectural components. A) Difference of Gaussian implements a center-surround receptive field. B) Local receptive fields of two neurons without weight sharing. C) Tuned divisive normalization inhibits each feature map by a Gaussian-weighted average of competing features. D) Log-polar transform simulating cortical magnification

Center-surround RFs are a common component of classical neuroscience models [31–33], where they are typically implemented using a Difference of Gaussian (DoG) that produces an excitatory peak at the RF center with an inhibitory surround (Fig. 1A). Although deep CNNs have the capacity to learn center-surround antagonism, supplementing traditional convolutional kernels with fixed-weight DoG kernels has been demonstrated to improve object recognition in the context of varied lighting, occlusion, and noise [34, 35].

**Local Receptive Fields**   The composition of convolutional operations in CNNs enables hierarchical processing and translation equivariance, both of which are fundamental to core object recognition in the primate ventral visual stream. However, the underlying mechanism through which this is achieved is biologically implausible, as kernel weights are shared among downstream neurons. Though locally connected neural network layers can theoretically learn the same operation, traditional convolutions are typically favored in practice for their computational efficiency and performance benefits. However, local connectivity is a ubiquitous pattern in the ventral stream (Fig. 1B), and visual processing phenomena (e.g., orientation preference maps [36]) have been attributed to this circuitry pattern. In artificial neural systems, Lee *et al.* [37] observed the emergence of topographic hallmarks in the inferior temporal cortex when encouraging local connectivity in CNNs. Pogodin *et al.* [38] considered the biological implausibility of CNNs and demonstrated a neuro-inspired approach to reducing the performance gap between traditional CNNs and locally-connected networks, meanwhile achieving better alignment with neural activity in primates.

**Divisive Normalization**   Divisive normalization is wide-spread across neural systems and species [39]. In early visual cortex, it is theorized to give rise to well-documented physiological phenomena, such as response saturation, sublinear summation of stimulus responses, and cross-orientation suppression [40].

In 2021, Burg and colleagues [41] introduced an image-computable divisive normalization model in which each artificial neuron was normalized by weighted responses of neurons with the same receptive field. In comparison to a simple 3-layer CNN trained to predict the same stimulus responses, their analyses revealed that cross-orientation suppression was more prevalent in the divisive normalization model than in the CNN, suggesting that divisive normalization may not be inherently learned by task-driven CNNs. In a separate study, Cirincione *et al.* [42] showed that simulating divisive normalization within a CNN can improve object recognition robustness to image corruptions and enhance alignment with certain tuning properties of primate V1.

**Tuned Normalization/Cross-Channel Inhibition**   While it is not entirely clear whether divisive normalization should be performed across space and/or across channels in computational models (implementations vary widely), Rust *et al.* [43] demonstrated that many response properties of motion-selective cells in the middle temporal area, such as motion-opponent suppression and response normalization, emerge from a mechanism they termed "tuned normalization". In this scheme, a given neuron is normalized by a pool of neurons that share the same receptive field but occupy a different region in feature space. We adopt this idea in the present work (Fig. 1C), hypothesizing that enforcing feature-specific weights in the pooling signal might enable a deep net to learn "opponent suppression" signals, much like cross-orientation signals found in biological V1 [44, 45].

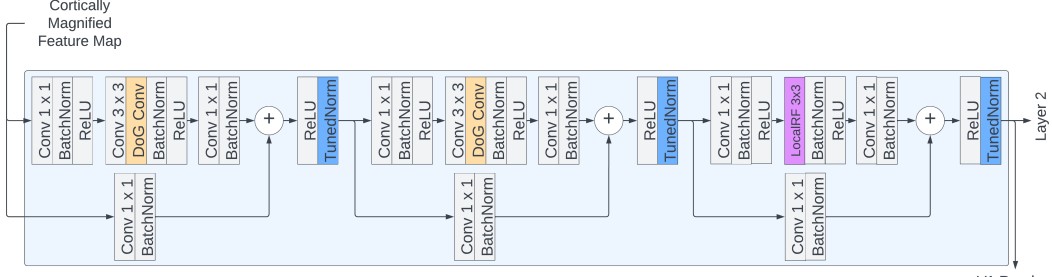

Figure 2: ResNet50 layer 1, supplemented with neuro-constrained architectural components. Throughout the the modified layer 1, primary visual cortex (V1) activity is modeled with cortical magnification, center-surround convolutions, tuned normalization, and local receptive field layers. Layer 1 output units are treated as artificial V1 neurons.

**Cortical Magnification** In many sensory systems, a disproportionately large area of the cortex is dedicated to processing the most important information. This phenomenon, known as cortical magnification, reflects the degree to which the brain dedicates resources to processing sensory information accompanying a specific sense. In the primary visual cortex, a larger proportion of cortical area processes visual stimuli presented at the center of the visual field as compared to stimuli at greater spatial eccentricities [46]. The relationship between locations in the visual field and corresponding processing regions in the visual cortex has commonly been modeled with a log-polar mapping (Fig. 1D) or derivations thereof [47–50].

Layers of artificial neurons of traditional CNNs have uniform receptive field sizes and do not exhibit any sort of cortical magnification, failing to capture these distinctive properties of neuronal organization in the primary visual cortex. Recent works have demonstrated that introducing log polar-space sampling into CNNs can give rise to improved invariance and equivariance to spatial transformations [51, 52] and adversarial robustness [53].

## 3 Methods

### 3.1 Neuro-Constrained CNN Architecture

Given the previous state-of-the-art V1 alignment scores achieved with ResNet50 [25], we adopted this architecture as our baseline and test platform. However, the architectural components that we considered in the work are modular and can be integrated into general CNNs architectures. The remainder of this subsection details the implementation and integration of each architectural component within a neuro-constrained ResNet. In all experiments, we treated the output units from ResNet50 layer 1 as "artificial V1" neurons (refer to Section 3.2 for layer selection criteria). Fig. 2 depicts ResNet50 layer 1 after enhancement with neuroscience-based architectural components. Code and materials required to reproduce the presented work are available at `github.com/bionicvisionlab/2023-Pogoncheff-Explaining-V1-Properties`.

**Center-Surround Antagonism** Center-surround ANN layers are composed of DoG kernels of shape $(c_i \times c_o \times k \times k)$, where $c_i$ and $c_o$ denote the number of input and output channels, respectively, and $k$ reflects the height and width of each kernel. These DoG kernels (Fig. 1A) are convolved with the pre-activation output of a standard convolution. Each DoG kernel, $DoG_i$ is of the form

$$\text{DoG}_i(x, y) = \frac{\alpha}{2\pi\sigma_{i,\text{center}}^2} \exp\left(-\frac{x^2 + y^2}{2\sigma_{i,\text{center}}^2}\right) - \frac{\alpha}{2\pi\sigma_{i,\text{surround}}^2} \exp\left(-\frac{x^2 + y^2}{2\sigma_{i,\text{surround}}^2}\right), \quad (1)$$

where $\sigma_{i,\text{center}}$ and $\sigma_{i,\text{surround}}$ were the Gaussian widths of the center and surround, respectively ($\sigma_{i,\text{center}} < \sigma_{i,\text{surround}}$), $\alpha$ was a scaling factor, and $(x, y) := (0, 0)$ at the kernel center. For $\alpha > 0$ the kernel will have an excitatory center and inhibitory surround, while $\alpha < 0$ results in a kernel with inhibitory center and excitatory surround. Novel to this implementation, each DoG kernel has learnable parameters, better accommodating the diverse tuning properties of neurons within the network. As in [34, 35], these DoG convolutions were only applied to a fraction of the input feature

map. Specifically, we applied this center-surround convolution to one quarter of all $3 \times 3$ convolutions in layer 1 of our neuro-constrained ResNet50.

**Local Receptive Fields**  To untangle the effects of local connectivity on brain alignment, we modified the artificial V1 layer by substituting the final $3 \times 3$ convolution of ResNet50 layer 1 with a $3 \times 3$ locally connected layer in isolation. This substitution assigns each downstream neuron its own filter while preserving its connection to upstream neurons (Fig. 1B), following the pattern in [38].

**Divisive Normalization**  We consider the divisive normalization block proposed in [42] which performs normalization both spatially and across feature maps using learned normalization pools. Following our experimental design principle of selectively modifying the network in the vicinity of the artificial V1 neurons, we added this divisive normalization block after the non-linear activation of each residual block in ResNet50 layer 1.

**Tuned Normalization**  We devised a novel implementation of tuned normalization inspired by models of opponent suppression [31, 43, 44]. In this scheme, a given neuron is normalized by a pool of neurons that share the same receptive field but occupy a different region in feature space (Fig. 1C), as in [41, 42]. Unlike the learned, weighted normalization proposed in [41], tuned inhibition was encouraged in our implementation by enforcing that each neuron was maximally suppressed by a neuron in a different region of feature space, and that no other neuron is maximally inhibited by activity in this feature space. Letting $x^c_{i,j}$ denote the activity of the neuron at spatial location $(i, j)$ and channel $c \in [1, C]$ after application of a non-linear activation function. The post-normalization state of this neuron, $x'^c_{i,j}$, is given by:

$$x'^c_{i,j} = \frac{x^c_{i,j}}{1 + \sum_k p_k x^{c_k}_{i,j}}, \tag{2}$$

where $p_{c,1}, \ldots, p_{c,C}$ defines a Gaussian distribution with variance $\sigma^2_c$ centered at channel $(c + \frac{C}{2}) \bmod C$. By defining $\sigma^2_c$ as a trainable parameter, task-driven training would optimize whether each neuron should be normalized acutely or broadly across the feature space.

As this mechanism preserves the dimension of the input feature map, it can follow any non-linear activation function of the core network without further modification to the architecture. Similar to the divisive normalization block, tuned normalization was added after the non-linear activation of each residual block in ResNet50 layer 1 in our experiments.

**Cortical Magnification**  Cortical magnification and non-uniform receptive field sampling was simulated in CNNs using a differentiable polar sampling module (Fig. 1D). In this module, the spatial dimension of an input feature map are divided into polar regions defined by discrete radial and angular divisions of polar space. In particular, we defined a discrete polar coordinate system partitioned in the first dimension by radial partitions $r_0, r_1, ..., r_m$ and along the second dimension by angular partitions $\theta_0, \theta_1, ..., \theta_n$. Pixels of the input feature map that are located within the same polar region (i.e., are within the same radial bounds $[r_i, r_{i+1})$ and angular bounds $[\theta_j, \theta_{j+1})$) are pooled and mapped to coordinate $(i, j)$ of the original pixel space (Fig. 1D) [54]. Pixels in the output feature map with no associated polar region were replaced with interpolated pixel values from the same radial bin. By defining the spacing between each concentric radial bin to be monotonically increasing (i.e., for all $i \in [1, m-1]$, $(r_i - r_{i-1}) \leq (r_{i+1} - r_i)$), visual information at lower spatial eccentricities with respect to the center of the input feature map consumes a larger proportion of the transformed feature map than information at greater eccentricities (Fig. F.1).

A notable result of this transformation is that any standard 2D convolution, with a kernel of size $k \times k$, that is applied to the the transformed features space is equivalent to performing a convolution in which the kernel covers a $k \times k$ contiguous region of polar space and strides along the angular and radial axes. Furthermore, downstream artificial neurons which process information at greater spatial eccentricities obtain larger receptive fields. Treating the CNN as a model of the ventral visual stream, this polar transformation immediately preceded ResNet50 layer 1 (replacing the first max-pooling layer), where V1 representations were assumed to be learned.

### 3.2 Training and Evaluation

**Training Procedure**   V1 alignment was evaluated for ImageNet-trained models [55]. For all models, training and validation images were downsampled to a resolution of $64 \times 64$ in consideration of computational constraints. Each model of this evaluation was randomly initialized and trained for 100 epochs with an initial learning rate of $0.1$ (reduced by a factor of $10$ at epochs $60$ and $80$, where validation set performance was typically observed to plateau) and a batch size of $128$.

We additionally benchmarked each neuro-constrained model on the Tiny-ImageNet-C dataset to study the effect of V1 alignment on object recognition robustness [56] (evaluation details provided in Appendix H). Tiny-ImageNet-C was used as an alternative to ImageNet-C given that the models trained here expected $64 \times 64$ input images and downsampling the corrupted images of ImageNet-C would have biased our evaluations. ImageNet pre-trained models were fine-tuned on Tiny-ImageNet prior to this evaluation. As a given model will learn alternative representations when trained on different datasets (thereby resulting in V1 alignment differences), we methodologically froze all parameters of each ImageNet trained model, with the exception of the classification head, prior to 40 epochs of fine tuning with a learning rate of $0.01$ and a batch size of $128$.

Validation loss and accuracy were monitored during both training procedures. The model state that enabled the greatest validation accuracy during training was restored for evaluations that followed. Training data augmentations were limited to horizontal flipping (ImageNet and Tiny-ImageNet) and random cropping (ImageNet).

Training was performed using single NVIDIA 3090 and A100 GPUs. Each model took approximately 12 hours to train on ImageNet and less than 30 minutes to fine-tune on Tiny-ImageNet.

**Evaluating V1 Alignment**   We evaluated the similarity between neuro-constrained models of V1 and the primate primary visual cortex using the Brain-Score V1 benchmark [16]. The V1 benchmark score is an average of two sub-metrics: 'V1 FreemanZiemba2013' and 'V1 Marques2020', which we refer to as V1 Predictivity and V1 Property scores in what follows. For each metric, the activity of artificial neurons in a given neural network layer is computed using in-silico neurophysiology experiments. The V1 Predictivity score reflects the degree to which the model can explain the variance in stimulus-driven responses of V1 neurons, as determined by partial least squares regression mapping. The V1 Property score measures how closely the distribution of 22 different neural properties, from 7 neural tuning categories (orientation, spatial frequency, response selectivity, receptive field size, surround modulation, texture modulation, and response magnitude), matches between the model's artificial neural responses and empirical data from macaque V1. Together, these two scores provide a comprehensive view of stimulus response similarity between artificial and primate V1 neurons.

Brain-Score evaluations assume a defined mapping between units of an ANN layer and a given brain region. In all analyses of V1 alignment that follow, we systematically fixed the output neurons of ResNet50 layer 1 as the artificial V1 neurons. Note that this is a more strict rule than most models submitted to the Brain-Score leaderboard, as researchers are able to choose which layer in the deep net should correspond to the V1 readout. In baseline analyses, among multiple evaluated layers, we observed highest V1 alignment between artificial units and primate V1 activity from layer 1, establishing it as a strong baseline. Alternative layer V1 scores are presented in Appendix B.

## 4   Results

### 4.1   Architectural Components in Isolation

Patterns of neural activity observed in the brain can be attributed to the interplay of multiple specialized processes. Through an isolated analysis, our initial investigations revealed the contribution of specialized mechanisms to explaining patterns of neural activity in V1. Tables 1 and 2 present the results of this analysis, including ImageNet validation accuracy, V1 Overall, V1 Predictivity, and V1 Property scores.

Among the five modules evaluated in this analysis, cortical magnification emerged as the most influential factor in enhancing V1 alignment. This mechanism substantially improved the ResNet's ability to explain the variance in stimulus responses, and the artificial neurons exhibited tuning properties that were more closely aligned with those of biological neurons, particularly in terms of

| | ImageNet Acc | V1 Overall | V1 Predictivity | V1 Property |
|---|---|---|---|---|
| Center-surround antagonism | .610 ± .001 | .545 ± .002 | .304 ± .016 | .786 ± .018 |
| Local receptive fields | .609 ± .001 | .550 ± .006 | .300 ± .002 | .799 ± .012 |
| Divisive normalization | .606 ± .001 | .543 ± .003 | .271 ± .014 | .815 ± .011 |
| Tuned normalization | .608 ± .002 | .547 ± .004 | .274 ± .004 | .820 ± .009 |
| Cortical magnification | .548 ± .008 | .587 ± .014 | .370 ± .008 | .805 ± .021 |
| ResNet50 (Baseline) | .613 ± .002 | .550 ± .004 | .295 ± .003 | .805 ± .011 |

Table 1: ImageNet object recognition classification performance ($64 \times 64$ images) and primary visual cortex (V1) alignment scores of ResNet50 augmented with each architectural component. Mean and standard deviations are reported across three runs (random initialization, training, and evaluating) of each architecture. Scores higher than baseline are presented in green and those lower are presented in red (the more saturated the color is, the greater the difference from baseline).

| | Orientation | Spatial frequency | Response selectivity | RF size | Surround modulation | Texture modulation | Response magnitude |
|---|---|---|---|---|---|---|---|
| Center-surround | .876 ± .027 | .831 ± .030 | .632 ± .012 | .853 ± .046 | .773 ± .027 | .757 ± .025 | .783 ± .024 |
| Local receptive fields | .904 ± .021 | .817 ± .016 | .648 ± .008 | .852 ± .054 | .847 ± .083 | .743 ± .036 | .780 ± .022 |
| Divisive normalization | .908 ± .017 | .840 ± .014 | .689 ± .007 | .858 ± .046 | .860 ± .070 | .746 ± .030 | .846 ± .019 |
| Tuned normalization | .907 ± .035 | .841 ± .013 | .689 ± .023 | .865 ± .031 | .852 ± .020 | .742 ± .029 | .844 ± .015 |
| Cortical magnification | .907 ± .037 | .848 ± .039 | .708 ± .011 | .803 ± .044 | .664 ± .063 | .789 ± .058 | .917 ± .071 |
| ResNet50 (Baseline) | .893 ± .023 | .826 ± .048 | .684 ± .059 | .832 ± .080 | .820 ± .009 | .786 ± .058 | .790 ± .042 |

Table 2: Model alignment across the seven primary visual cortex (V1) tuning properties that constitute the V1 Property score. Mean and standard deviation of scores observed across three trials of model training and evaluation are reported.

orientation tuning, spatial frequency tuning, response selectivity, and most of all, stimulus response magnitude. However, the artificial neuronal responses of the cortical magnification network showed lower resemblance to those observed in primate V1 with regard to surround modulation, as compared to the baseline network.

Simulating neural normalization within the ResNet resulted in artificial neurons that displayed improved alignment with primate V1 in terms of response properties. Noteworthy enhancements were observed in the spatial frequency, receptive field size, surround modulation, and response magnitude properties of neurons within the modified network, leading to improvements in the V1 Property score. These results applied to both tuned and untuned forms of normalization.

In contrast, the introduction of center-surround convolutions yielded minimal improvements in neural predictivity and slight reductions in overall neuron property similarity. Surprisingly, the surround modulation properties of the artificial neurons decreased compared to the baseline model, contrary to our expectations.

Finally, replacing the final $3 \times 3$ convolution preceding the artificial V1 readout with a locally connected layer resulted in modest changes in V1 alignment. This was one of the two mechanisms that led to improvements in the surround modulation response property score (tuned normalization being the other).

These findings collectively provide valuable insights into the individual contributions of each specialized mechanism. Although mechanisms simulating center-surround antagonism (i.e., DoG convolution) and local connectivity provide little benefit to overall predictivity and property scores in isolation, we observed that they reduce the property dissimilarity gap among tuning properties that are nonetheless important and complement alignment scores where divisive normalization and cortical magnification do not.

## 4.2 Complementary Components Explain V1 Activity

Constraining a general-purpose deep learning model with a single architectural component is likely insufficient to explain primate V1 activity given our knowledge that a composition of known circuits play pivotal roles in visual processing. This was empirically observed in Section 4.1, wherein cortical

| Center-Surround | Local RF | Tuned Normalization | Cortical Magnification | ImageNet Acc | V1 Overall | V1 Predictivity | V1 Property |
|:---:|:---:|:---:|:---:|---:|---:|---:|---:|
| ✓ | ✓ | ✓ | ✓ | .551 | **.605** | .357 | .852 |
|  | ✓ | ✓ | ✓ | .543 | **.605** | .353 | .857 |
| ✓ |  | ✓ | ✓ | .541 | .599 | .340 | **.858** |
| ✓ | ✓ |  | ✓ | .552 | .592 | .364 | .820 |
| ✓ | ✓ | ✓ |  | .603 | .555 | .276 | .834 |
|  |  | ✓ | ✓ | .541 | .598 | .351 | .845 |
|  | ✓ |  | ✓ | .555 | .593 | **.384** | .803 |
|  | ✓ | ✓ |  | .606 | .561 | .287 | .835 |
| ResNet50 (Baseline) | | | | **.613** | .550 | .295 | .805 |

Table 3: Backward component elimination results. Checkmarks denote whether or not the architectural component was included in the model. Adversarial training was performed on the two models that tied for the top V1 Overall Score.

magnification was the only architectural component found to improve the overall V1 alignment score. Taking inspiration from this design principle, we supplemented a ResNet50 with each architectural component and discern the necessary components to achieve optimal V1 alignment in an ablation study. We omitted the architectural component implementing divisive normalization, however, as it it cannot be integrated simultaneously with tuned normalization, which was observed to yield slightly higher V1 Predictivity and Property scores in isolated component evaluation. Starting with a ResNet which features all of these architectural components, we employed a greedy approach reminiscent of backward elimination feature selection to deduce critical components without having to evaluate every permutation of component subsets. In each round of this iterative approach, we selectively removed the architectural component that reduced overall V1 alignment the most until only one feature remained. This analysis allowed us to identify the subset of components that collectively yielded the most significant improvements in V1 alignment, and unraveled the intricate relationship between these specialized features and their combined explanation of V1.

The results of the ablation study are presented in Table 3. With the exception of center-surround antagonism, removing any neural mechanisms from the modified residual network reduced overall V1 alignment, suggesting that (1) each architectural component contributed to V1 alignment (the utility of center-surround antagonism is detailed in Section 4.5) and (2) nontrivial interactions between these mechanisms explain V1 more than what is possible with any single mechanism. Seven of the eight models evaluated in this ablation study substantially outperformed all existing models on the Brain-Score platform in modeling V1 tuning property distributions. Furthermore, four models were observed to achieve state-of-the-art V1 Overall scores, explaining both V1 stimulus response activity and neural response properties with high fidelity.

Whether or not feed-forward, ImageNet-trained ANNs can fully approximate activity in primate V1 has stood as an open question. Previous studies have argued that no current model is capable of explaining all behavioral properties using neurons from a single readout layer [17]. The top performing models of the current evaluation stand out as the first examples of CNNs with neural representations that accurately approximate all evaluated V1 tuning properties (Appendix C), offering positive evidence for the efficacy of explaining primate V1 with neuro-inspired deep learning architectures.

## 4.3 Component Contributions to V1 Alignment

Supplementing the quantitative studies of Sections 4.1 and 4.2, studying learned component parameters and visual stimuli that maximally activated artificial V1 neurons provided further insights into the role of these architectural components in explaining biological V1 activity.

The V1 Property scores found in our ablation studies (Table 5) suggest the contribution of center-surround antagonism in aligning spatial frequency properties between artificial and biological V1 representations. Optimized DoG kernels implementing center-surround antagonism were commonly observed to have center Gaussian distributions with low variance (i.e., small $\sigma^2_{center}$), suggesting preferences for patterns and textures with high spatial frequencies (Appendix Fig. F.2).

| Center-Surround | Local RF | Tuned Nor-malization | Cortical Mag-nification | Adversarial Training | ImageNet Acc | V1 Overall | V1 Predictivity | V1 Property |
|---|---|---|---|---|---|---|---|---|
| ✓ | ✓ | ✓ | ✓ | ✓ | .448 | **.629** | **.430** | .829 |
| | ✓ | ✓ | ✓ | ✓ | .448 | .625 | **.430** | .819 |
| Adversarially trained ResNet50 (Baseline) | | | | | **.555** | .581 | .352 | .809 |

Table 4: Adversarial training was performed on the two models that tied for the top V1 Overall Score. Checkmarks denote whether or not the architectural component was included in the model.

Regarding local receptive fields, we hypothesized that removing weight sharing would enable a greater diversity of response selectivity patterns to be learned. While multi-component ablation studies revealed a reduction in response selectivity property scores when local filtering was omitted, the same finding was surprisingly not observed in the single-component analyses of Section 4.1.

Given the inter-neuron competition enforced by tuned normalization, one would expect networks with this component to learn more diverse artificial V1 representations. Analyses of the visual stimuli that maximally activated artificial neurons of these networks (i.e., optimized visual inputs that maximally excite neurons of each channel of the artificial V1 layer, computed via stochastic gradient ascent) provide evidence for this. Quantitatively, we found that the mean Learned Perceptual Image Patch Similarity (LPIPS) [57] between pairs of maximally activating stimuli of the tuned normalization network was less than that of the baseline ResNet50 network ($p < 0.01$, one-way ANOVA [58]). We suggest that this learned feature diversity contributed to the improvements in spatial frequency, response selectivity, receptive field size, surround modulation, and response magnitude tuning property scores when tuned normalization was present.

Finally, given the retinotopic organization of V1, we hypothesized that cortical magnification would give rise to better-aligned response selectivity and receptive field size tuning distributions, meanwhile improving V1 neuron predictivity. In each trial of our ablation studies for which cortical magnification was removed, these respective scores dropped, supporting this hypothesis.

## 4.4 Object Recognition Robustness to Corrupted Images

In contrast with the human visual system, typical CNNs generalize poorly to out-of-distribution data. Small perturbations to an image can cause a model to output drastically different predictions than it would on the in-tact image. Recent studies have demonstrated a positive correlation between model-brain similarity and robustness to image corruptions [24–26, 42, 59] After fine-tuning each model's classification head on Tiny-ImageNet (see Section 3.2), we evaluated the object recognition accuracy of each model from Section 4.1 and the top two overall models from Section 4.2 on the Tiny-ImageNet-C dataset. The results of these evaluations for each category of corruption and corruption strength are provided in Appendix H.

Among the evaluated components, only tuned normalization was observed to yield improved corrupt image classification accuracy over the entire test set, albeit slight, beating the baseline accuracy (0.278) by 0.005 (i.e., an absolute improvement of .5%). More substantial improvements were observed on 'brightness', 'defocus_blur', 'elastic_transform', and 'pixelate' corruptions (improvements over the baseline of .00986, .00989, .0105, and .0133, respectively).

## 4.5 Adversarially Training Neuro-Constrained ResNets

Adversarial training has previously been shown to enhance the brain-similarity of artificial neural representations without any modification to the underlying network [25, 60]. Curious as to whether adversarial training would further align the neuro-constrained ResNet50s with V1 activity, we selectively trained the two networks most aligned with V1 (one model with all architectural components and the other with all components except center-surround convolution) from Section 4.2 using "Free" adversarial training [61] (Appendix G). The results are shown in Table 4. Despite the drop in object recognition accuracy, the artificial neural representations that emerged in each network were drastically better predictors of stimulus response variance representations. Tuning property alignment dropped in the process, but remained above previous state-of-the-art regardless. Interestingly, we found that the main difference in V1 scores between these two models can be traced to surround modulation tuning alignment. Center-surround convolutions indeed contributed to improved surround

modulation tuning learned while training with on corrupted images, contrasting its apparent lack of contribution to the overall network suggested in the ablation study.

In sum, both networks achieved Rank-1 V1 Overall, Predictivity, and Property scores by large margins, setting a new standard in this breed of brain-aligned CNNs. At the time of writing, the previous Rank-1 V1 Overall, Predictivity, and Property scores were .594, .409, and .816, respectively, and all achieved by separate models.

## 5 Discussion

Throughout this work we presented a systematic evaluation of four architectural components derived from neuroscience principles and their influence on model-V1 similarity. Specifically, we studied task-driven CNNs enhanced with ANN layers that simulate principle processing mechanisms of the primate visual system including center-surround antagonism, local receptive fields, tuned normalization, and cortical magnification. Through an ablation study and isolated component analyses, we found that each component contributed to the production of latent ANN representations that better resemble those of primate V1, as compared to a traditional baseline CNN. When these four components were assembled together within a neuro-constrained ResNet50, V1 tuning properties were explained better than any previous deep learning model that we are aware of. Furthermore, this neuro-constrained model exhibited state-of-the-art explanation of V1 neural activity and is the first of its kind to do so, by a large margin nonetheless, highlighting a promising direction in biologically constrained ANNs. Training this model with "free" adversarial training greatly improved its ability to predict primate neural response to image stimuli at a minor sacrifice to tuning property similarity, establishing an even larger gap between previous state of the art.

Among all architectural components examined in this work, cortical magnification was the most influential to improving V1 alignment. This mechanism on its own could not explain the neural activity as comprehensively as the top models of this study, however. Our implementation of tuned normalization provided substantial improvement to V1 tuning property alignment, and was the only component that contributed to model robustness. The importance of center-surround antagonism seemed to be training data-dependent. In our ablation study, for which all models were trained on ImageNet, center-surround convolutional layers did not contribute to overall V1 scores. This did not surprise us, as deep CNNs have the capacity to learn similar representations without these specialized layers. When training on adversarially perturbed data, however, the center-surround antagonism provided by this layer appeared to improve surround modulation tuning properties of artificial V1 neurons. While previous attempts at improving model-brain similarity have been highly dataset dependent, our results highlight the importance of artificial network design.

A notable limitation to our work is the reduction in ImageNet classification performance that was observed upon the introduction of cortical magnification. While maintaining baseline model accuracy was not a motivation of this work, we can imagine situations in which object recognition performance needs to be preserved alongside these improvements in brain-model alignment. The implementation of cortical magnification in this work assumed that the model's gaze was focused at the center of each image. Consequently, visual stimuli at greater eccentricities from the image center were under-sampled during the polar transformation (Fig. F.1), making images for which the object of interest was not located at the image center (common in ImageNet) more challenging to classify. One scope of future work involves implementing saliency- or attention-driven polar transformations that dynamically focus the center of the polar map on, or near, an object of interest as opposed to being fixed at the image center. We demonstrate the efficacy of this strategy with a naïve proof of concept in which classification is performed by averaging predictions from the five static crops of each image [62]. This simple strategy improved the validation accuracy of the network with all components from 55.1% to 59.9% without affecting V1 scores. A more sophisticated, dynamic strategy could further reduce this accuracy drop. We additionally plan to extend this work to model architectures other than ResNet to validate the widespread application of each of these neuro-constrained components.

This work highlights an important advancement in the field of NeuroAI, as we systematically establish a set of neuro-constrained architectural components that contribute to state-of-the-art V1 alignment. We argue that our architecture-driven approach can be further generalized to additional areas of the brain as well. The neuroscience insights that could be gleaned from increasingly accurate in-silico models of the brain have the potential to transform the fields of both neuroscience and AI.

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

# Appendix

## A    Supplemental Model Diagrams

Fig. A.1 depicts the modifications made to ResNet50 residual layer 1 in the isolated component analyses of section 4.1. All multi-component (composite) models analyzed in Section 4.2 relied on combinations of these modifications (as exemplified in Fig. 2).

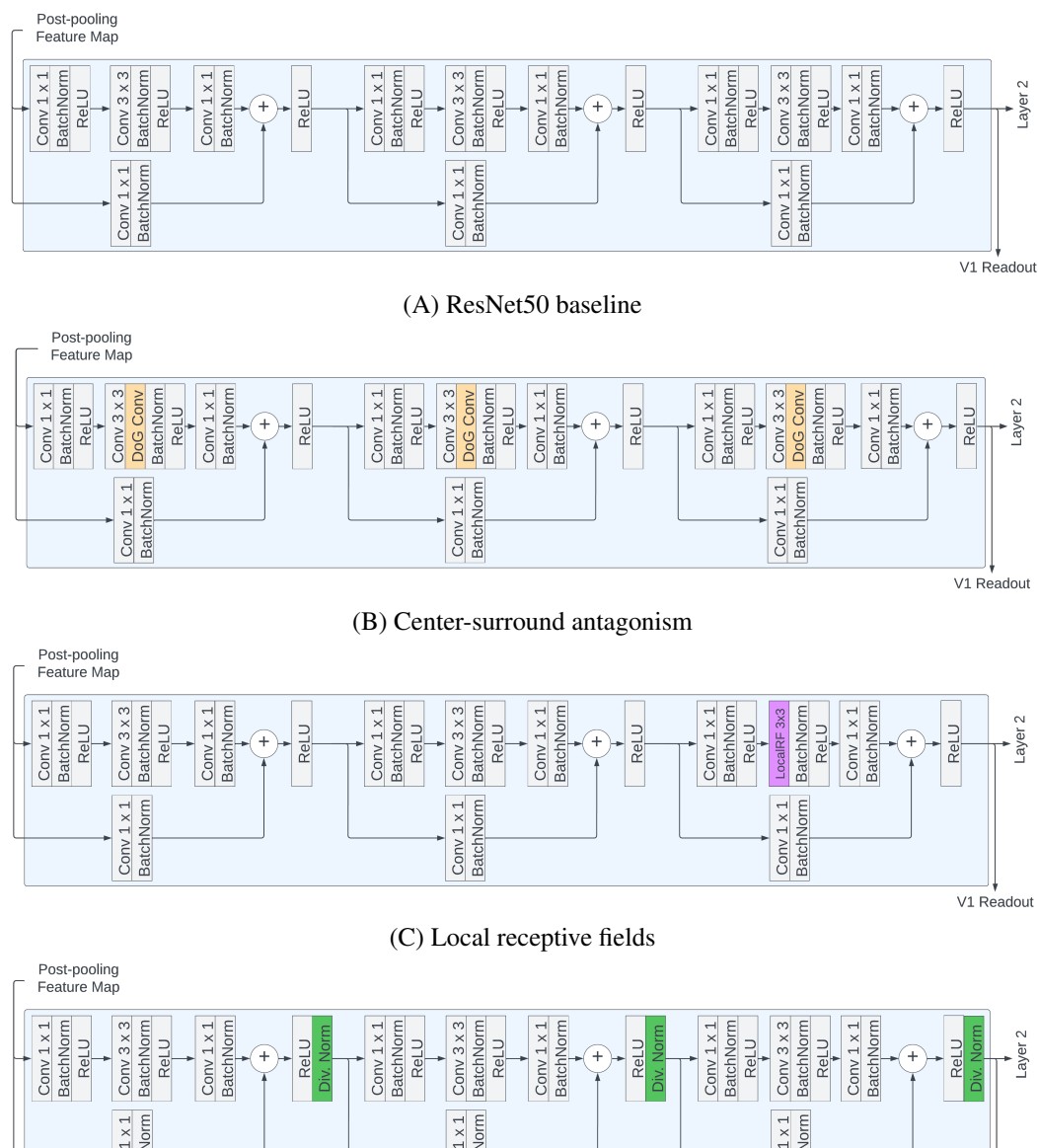

(A) ResNet50 baseline

(B) Center-surround antagonism

(C) Local receptive fields

(D) Divisive normalization

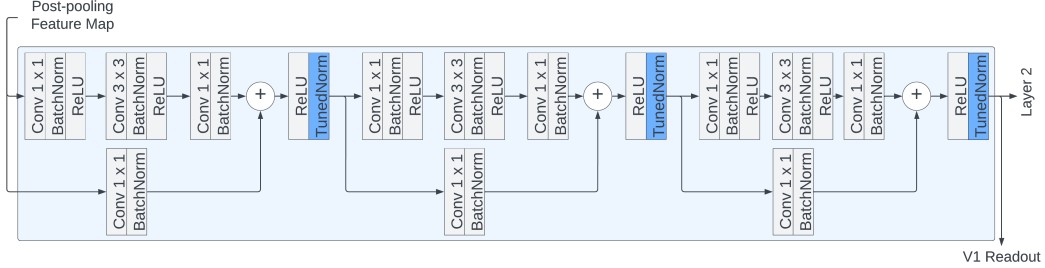

(E) Tuned normalization

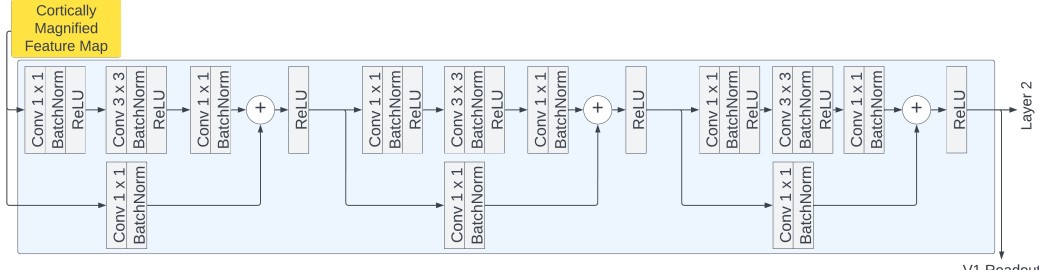

(F) Cortical magnification

Figure A.1: ResNet50 residual layer 1, supplemented with individual neuro-constrained architectural components, as in section 4.1. (A) No modification (baseline ResNet50 layer 1), (B) with center-surround antagonism, (C) with local receptive field (RF), (D) with divisive normalization, (E) with tuned normalization, (F) with cortical magnification.

# B  V1 Scores of Alternate Layers of Baseline Network

When evaluating a model on Brain-Score, users are permitted to commit a mapping between model layers and areas of the ventral stream. Model-brain alignment is computed for each mapped pair in the Brain-Score evaluation. To promote a fair evaluation, we sought to find the layer that yielded optimal V1 alignment from the baseline ResNet50 model and fix this layer as the artificial V1 readout layer in all of our tested models. It is worth noting that after supplementing the base ResNet50 with neuro-constrained components, this layer may no longer offer optimal V1 alignment in the augmented network. In spite of this, we maintain this layer as our artificial V1 readout layer for fair evaluation.

To find the ResNet50 layer with the best V1 Overall, Predictivity, and Property scores, we compared a total of 20 different hidden layers (Fig. B.1). 16 of these layers corresponded to the post-activation hidden states of the network. The remaining 4 were downsampling layers of the first bottleneck block of each residual layer in the network, as these have previously demonstrated good V1 alignment [25]. Aside from these downsampling layers, hidden layers that did not follow a ReLU activation were omitted from this evaluation as the activities of these states can take on negative values and are therefore less interpretable as neural activities. Among all evaluated layers, the final output of ResNet50 residual layer 1 (i.e., the output of the third residual block of ResNet50) offered the highest V1 Overall score, and was therefore selected as the artificial V1 readout layer in all of our experiments.

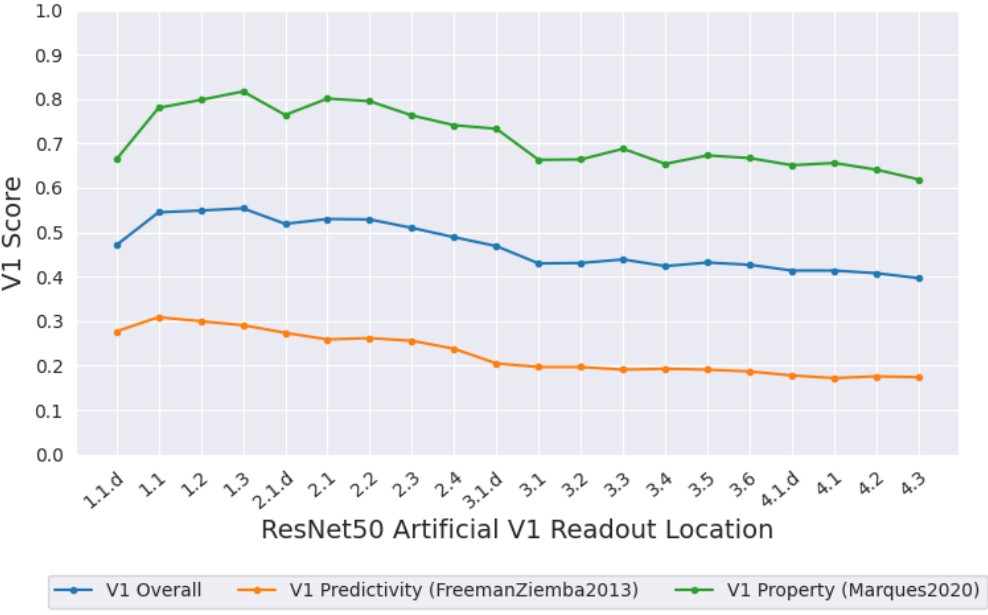

Figure B.1: V1 alignment Brain-Scores for 20 different hidden layers of ResNet50. In the plot above, readout location 'X.Y' denotes that artificial V1 activity was evaluated from residual block 'Y' of residual layer 'X'. Readout location suffixed with '.d' correspond to downsampling layers of the associated residual bottleneck. Highest V1 overall score came from block 3 of residual layer 1.

## C   Expanded Model Tuning Properties

Primary visual cortex (V1) tuning property alignments for each composite model evaluated in Section 4.2 are presented in Table 5. Tuning property similarities are computed as ceiled Kolmogorov-Smirnov distance between artificial neural response distributions from the model and empirical distributions recorded in primates [16, 17].

| Center-Surround | Local RF | Tuned Norm. | Cortical Mag. | Adv. Training | Orientation | Spatial Frequency | Response Selectivity | RF Size | Surround Modulation | Texture Modulation | Response Mag. |
|---|---|---|---|---|---|---|---|---|---|---|---|
| ✓ | ✓ | ✓ | ✓ |  | .891 | .925 | .756 | .840 | .779 | .844 | .930 |
|  | ✓ | ✓ | ✓ |  | .858 | .919 | .780 | .834 | .808 | .871 | .930 |
| ✓ |  | ✓ | ✓ |  | .894 | .932 | .750 | .851 | .775 | .858 | .946 |
| ✓ | ✓ |  | ✓ |  | .878 | .873 | .739 | .816 | .719 | .802 | .910 |
| ✓ | ✓ | ✓ |  |  | .875 | .873 | .702 | .808 | .890 | .815 | .870 |
|  | ✓ |  | ✓ |  | .873 | .886 | .735 | .840 | .794 | .825 | .959 |
| ✓ |  |  | ✓ |  | .902 | .866 | .715 | .801 | .625 | .841 | .869 |
|  | ✓ | ✓ |  |  | .915 | .817 | .691 | .811 | .898 | .802 | .911 |
| ✓ | ✓ | ✓ | ✓ | ✓ | .924 | .863 | .773 | .797 | .733 | .815 | .899 |
| ✓ | ✓ | ✓ | ✓ |  | .944 | .834 | .768 | .806 | .673 | .811 | .900 |

Table 5: Ablation study model alignment across the seven primary visual cortex (V1) tuning properties that constitute the V1 Property score ('Marques2020') of Brain-Score. Checkmarks denote whether or not the architectural component was included in the model.

## D   V1 Brain-Scores of Untrained Models

|  | V1 Overall | V1 Predictivity | V1 Property |
|---|---|---|---|
| Center-surround antagonism | .298 | .245 | .551 |
| Local receptive fields | .477 | .210 | .743 |
| Divisive normalization | .499 | .207 | .792 |
| Tuned normalization | .471 | .218 | .724 |
| Cortical magnification | .497 | .276 | .718 |
| All Components | .483 | .225 | .741 |
| ResNet50 | .466 | .223 | .710 |

Table 6: primary visual cortex (V1) alignment scores of untrained ResNet50 model variants.

## E   V2, V4, and IT Brain-Scores of Top Model

Table 7 shows the Brain-Scores of our top performing V1 model (the adversarially trained ResNet50 with all architectural components) for brain areas V2, V4, and IT. Network layers were mapped to visual areas V2, V4, and IT by finding the layers that achieve the best scores on these visual area benchmarks, as evaluated on Brain-Score's publicly available evaluation set.

| Visual Area Brain-Score | V2 | V4 | IT |
|---|---|---|---|
|  | .298 | .245 | .551 |

Table 7: V2, V4, and IT Brain-Scores of adversarially trained ResNet50 with all architectural components.

# F   Supplemental Visualizations

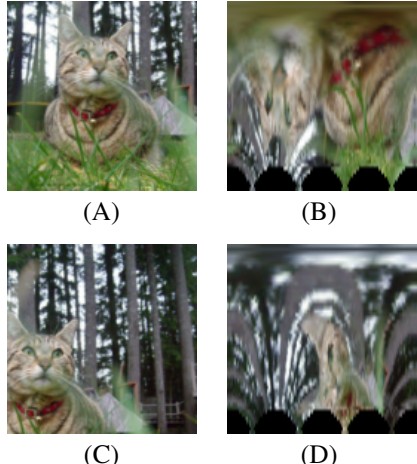

Figure F.1: Cortical magnification disproportionately samples the input image (or feature map). (A) Original image in which the object of interest (cat) is centered in the image frame. (B) Image (A) after the simulated cortical magnification transform. (C) Different crop of the original image, with the cat offset from image center. (D) Image (C) after the simulated cortical magnification transform

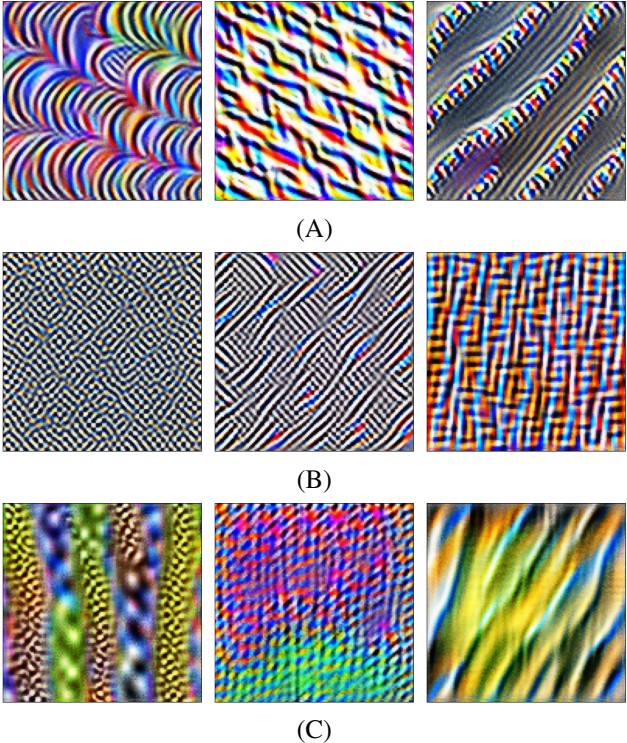

Figure F.2: Visual stimuli that maximally activate artificial V1 neurons of (A) baseline, (B) center-surround, and (C) tuned normalized ResNet50 networks. Maximally activating images generated using the python package 'lucent' (`https://https://github.com/greentfrapp/lucent`).

# G   Adversarial Training

The neuro-constrained ResNets discussed in Section 4.5 were trained using the "Free" adversarial training method proposed by Shafahi *et al.* [61]. In Projected Gradient Descent (PGD)-based adversarial training (a typical approach to adversarially training robust classifiers), a network is trained on adversarial samples that are generated on the fly during training. Specifically, in PGD-based adversarial training, a batch of adversarial images is first generated through a series of iterative perturbations to an original image batch, at which point the parameters of the network are finally updated according to the network's loss, as evaluated on the adversarial examples. "Free" adversarial training generates adversarial training images with a similar approach, but the parameters of the network are simultaneously updated with every iteration of image perturbation, significantly reducing training time. The authors refer to these mini-batch updates as "replays", and refer to the number of replays of each mini-batch with the parameter $m$.

The adversarially trained models of Section 4.5 were trained with $m = 4$ replays and perturbation clipping of $\epsilon = \frac{2}{255}$. These models were trained for $120$ epochs using a stochastic gradient descent optimizer with an initial learning rate of $0.1$, which was reduced by a factor of $10$ every $40$ epochs, momentum of $0.9$, and weight decay of $1 \times 10^{-5}$. Each model was initialized with the weights that were learned during traditional ImageNet training for the analyses in Section 4.2. "Free" adversarial training was performed using code provided by the authors of this method (`https://github.com/mahyarnajibi/FreeAdversarialTraining`).

# H   Robustness to Common Image Corruptions

## H.1   Dataset Description

We evaluated image classification robustness to common image corruptions using the Tiny-ImageNet-C dataset [56]. Recall that Tiny-ImageNet-C was used instead of ImageNet-C, because our models were trained on $64 \times 64$ input images. Downscaling ImageNet-C images would have potentially altered the intended corruptions and biased our evaluations.

Tiny-ImageNet-C is among a collection of corrupted datasets (e.g., ImageNet-C, CIFAR-10-C, CIFAR-100-C) that feature a diverse set of corruptions to typical benchmark datasets. Hendrycks and Dietterich [56] suggest that given the diversity of corruptions featured in these datasets, performance on these datasets can be seen as a general indicator of model robustness. The Tiny-ImageNet-C evaluation dataset consists of images from that Tiny-ImageNet validation dataset that have been corrupted according to $15$ types of image corruption, each of which is categorized as a 'noise', 'blur', 'weather', or 'digital' corruption. The $15$ corruption types include: Gaussian noise, shot noise, impulse noise, defocus blur, frosted glass blur, motion blur, zoom blur, snow, frost, fog, brightness, contrast, elastic transformation, pixelation, and JPEG compression. Each corruption is depicted in Fig. H.1. Every image of this evaluation dataset is also corrupted at five levels of severity (the higher the corruption severity, the more the original image had been corrupted). Corruption severities for Gaussian noise are exemplified in Fig. H.2.

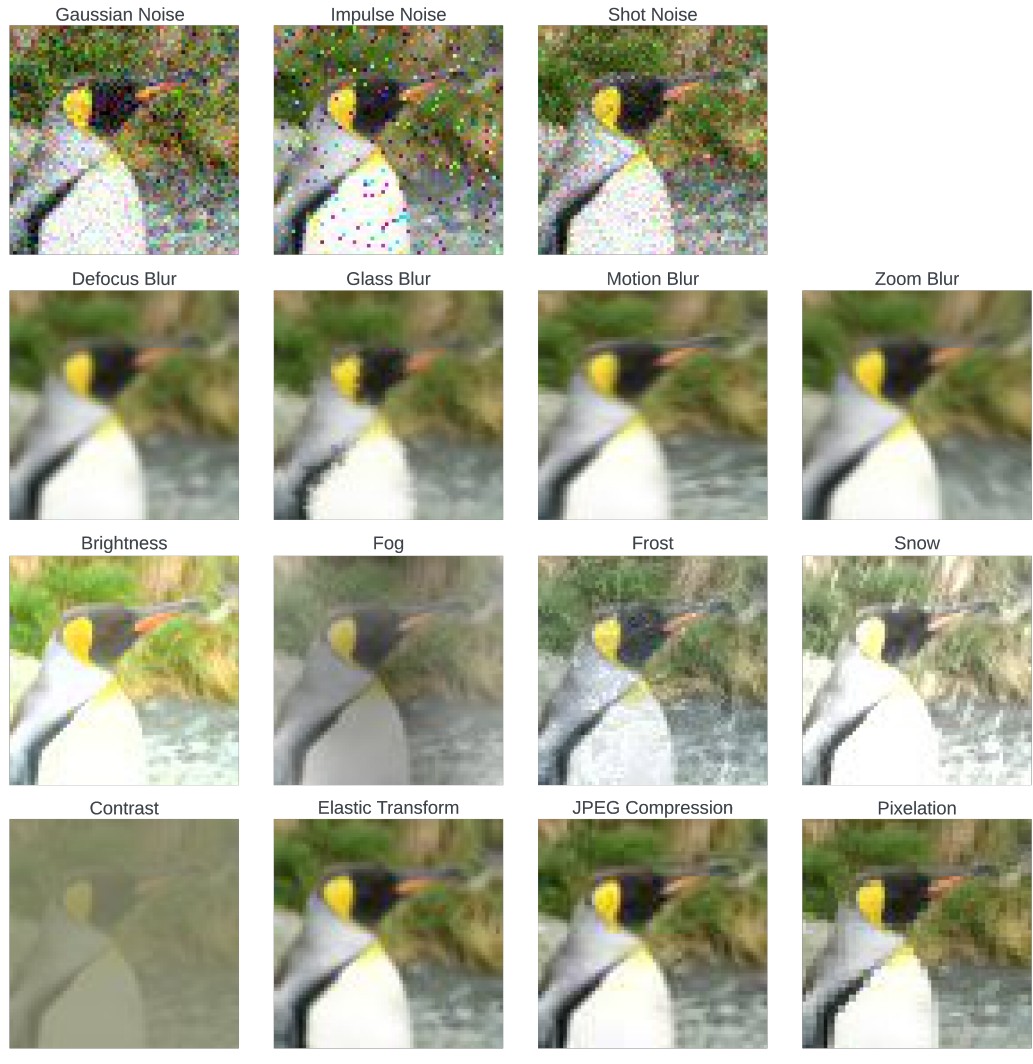

Figure H.1: 15 corruptions of the Tiny-ImageNet-C dataset, applied to a sample image from Tiny-ImageNet-C. First row: noise corruptions, second row: blur corruptions, third row: weather corruptions, bottom row: digital corruptions. All corruptions shown at severity level 3.

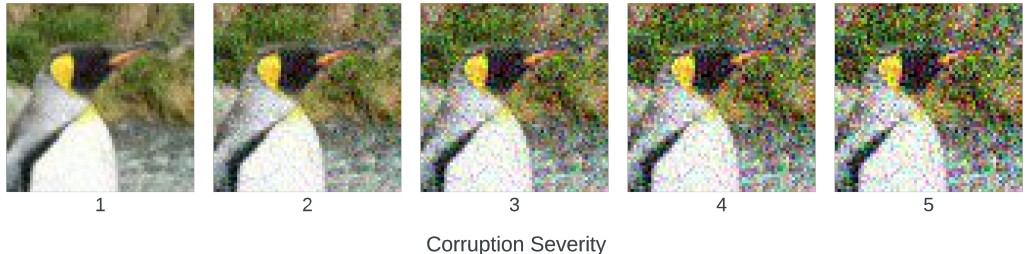

Corruption Severity

Figure H.2: Gaussian noise corruption, shown at corruption severity levels 1-5.

## H.2 Corrupted Image Robustness

A detailed breakdown of Tiny-ImageNet-C image classification accuracy for each single-component, neuro-constrained ResNet-50 and the composite models that achieved top V1 Overall score without adversarial training are provided in Tables 8, 9, and 10.

| | Tiny-ImageNet Val. | Tiny-ImageNet-C | $\Delta$ |
|---|---|---|---|
| ResNet50 (Baseline) | $\mathbf{.742} \pm .003$ | $.278 \pm .004$ | $.463 \pm .006$ |
| Center-surround antagonism | $.739 \pm .004$ | $.277 \pm .008$ | $.463 \pm .009$ |
| Local Receptive Fields | $.741 \pm .002$ | $.275 \pm .004$ | $.467 \pm .004$ |
| Tuned Normalization | $.740 \pm .001$ | $\mathbf{.283} \pm .006$ | $\mathbf{.457} \pm .006$ |
| Cortical Magnification | $.683 \pm .001$ | $.222 \pm .009$ | $.461 \pm .009$ |
| Composite Model A | .694 | .231 | .463 |
| Composite Model B | .691 | .232 | .459 |

Table 8: Classification accuracy of models on Tiny-ImageNet validation and Tiny-ImageNet-C (all corruption types and severities) datasets. Composite Model A includes all 4 neuro-constrained architectural components (center-surround antagonism, local receptive fields, tuned normalization, and cortical magnification). Composite Model B contained all architectural components, with the exception of center-surround antagonism. For baseline and single-component models, mean accuracies ($\pm$ one standard deviation) are reported, where each trial was associated with a distinct base model from the repeated trials of section 4.1.

| | Corruption Severity | | | | |
|---|---|---|---|---|---|
| | 1 | 2 | 3 | 4 | 5 |
| ResNet50 (Baseline) | $.418 \pm .004$ | $.345 \pm .005$ | $.269 \pm .004$ | $.204 \pm .005$ | $.156 \pm .003$ |
| Center-surround antagonism | $.414 \pm .010$ | $.343 \pm .009$ | $.267 \pm .009$ | $.203 \pm .006$ | $.156 \pm .004$ |
| Local Receptive Fields | $.416 \pm .003$ | $.341 \pm .003$ | $.264 \pm .003$ | $.199 \pm .002$ | $.153 \pm .002$ |
| Tuned Normalization | $\mathbf{.424} \pm .006$ | $\mathbf{.350} \pm .006$ | $\mathbf{.274} \pm .007$ | $\mathbf{.208} \pm .006$ | $\mathbf{.160} \pm .004$ |
| Cortical Magnification | $.349 \pm .011$ | $.277 \pm .013$ | $.208 \pm .010$ | $.157 \pm .007$ | $.120 \pm .006$ |
| Composite Model A | .363 | .289 | .216 | .163 | .125 |
| Composite Model B | .361 | .288 | .219 | .165 | .127 |

Table 9: Classification accuracy of models on Tiny-ImageNet-C at each level of corruption severity. Composite Model A includes all 4 neuro-constrained architectural components (center-surround antagonism, local receptive fields, tuned normalization, and cortical magnification). Composite Model B contained all architectural components, with the exception of center-surround antagonism. For baseline and single-component models, mean accuracies ($\pm$ one standard deviation) are reported, where each trial was associated with a distinct base model from the repeated trials of section 4.1.

|  | Noise Corruptions | | | |
|---|---|---|---|---|
|  | Gaussian Noise | Impulse Noise | Shot Noise | Avg. |
| ResNet50 (Baseline) | **.197** ± .011 | .191 ± .010 | **.232** ± .013 | **.207** ± .011 |
| Center-surround antagonism | .195 ± .010 | .186 ± .009 | **.232** ± .012 | .204 ± .010 |
| Local Receptive Fields | .185 ± .006 | .184 ± 009 | .219 ± .010 | .196 ± .008 |
| Tuned Normalization | .195 ± .008 | **.192** ± .004 | .228 ± .007 | .205 ± .006 |
| Cortical Magnification | .150 ± .008 | .157 ± .007 | .180 ± .011 | .162 ± .008 |
| Composite Model A | .151 | .156 | .184 | .164 |
| Composite Model B | .144 | .149 | .177 | .157 |

|  | Blur Corruptions | | | | |
|---|---|---|---|---|---|
|  | Defocus Blur | Glass Blur | Motion Blur | Zoom Blur | Avg. |
| ResNet50 (Baseline) | .224 ± .003 | .182 ± .001 | .272 ± .003 | .241 ± .004 | .230 ± .002 |
| Center-surround antagonism | .223 ± .009 | .184 ± .004 | .274 ± .012 | .243 ± .011 | .231 ± .009 |
| Local Receptive Fields | .228 ± .006 | .183 ± .004 | .273 ± .005 | .243 ± .008 | .232 ± .005 |
| Tuned Normalization | **.234** ± .009 | **.188** ± .002 | **.277** ± .009 | **.248** ± .010 | **.237** ± .007 |
| Cortical Magnification | .174 ± .010 | .162 ± .008 | .222 ± .007 | .190 ± .006 | .187 ± .008 |
| Composite Model A | .186 | .167 | .236 | .200 | .197 |
| Composite Model B | .196 | .174 | .249 | .222 | .210 |

|  | Weather Corruptions | | | | |
|---|---|---|---|---|---|
|  | Brightness | Fog | Frost | Snow | Avg. |
| ResNet50 (Baseline) | .401 ± .005 | **.282** ± .003 | .360 ± .006 | .310 ± .004 | .338 ± .004 |
| Center-surround antagonism | .399 ± .008 | .270 ± .008 | .357 ± .012 | .302 ± .003 | .332 ± .007 |
| Local Receptive Fields | .398 ± .008 | .275 ± .005 | .351 ± .006 | .298 ± .004 | .331 ± .003 |
| Tuned Normalization | **.410** ± .008 | **.282** ± .011 | **.361** ± .006 | **.311** ± .010 | **.341** ± .008 |
| Cortical Magnification | .327 ± .011 | .211 ± .013 | .283 ± .014 | .248 ± .010 | .267 ± .011 |
| Composite Model A | .338 | .220 | .286 | .258 | .275 |
| Composite Model B | .327 | .225 | .284 | .255 | .273 |

|  | Digital Corruptions | | | | |
|---|---|---|---|---|---|
|  | Contrast | Elastic | JPEG | Pixelate | Avg. |
| ResNet50 (Baseline) | .125 ± .001 | .331 ± .007 | .454 ± .007 | .374 ± .003 | .321 ± .003 |
| Center-surround antagonism | .122 ± .002 | .331 ± .014 | .455 ± .007 | .374 ± .004 | .321 ± .006 |
| Local Receptive Fields | .120 ± .004 | .329 ± .003 | .457 ± .005 | .375 ± .002 | .320 ± .001 |
| Tuned Normalization | **.128** ± .008 | **.342** ± .010 | **.463** ± .006 | **.387** ± .006 | **.330** ± .007 |
| Cortical Magnification | .082 ± .005 | .287 ± .007 | .374 ± .013 | .287 ± .014 | .257 ± .010 |
| Composite Model A | .081 | .305 | .397 | .303 | .272 |
| Composite Model B | .086 | .314 | .383 | .293 | .269 |

Table 10: Corrupted image classification accuracy by corruption type. Composite Model A includes all 4 neuro-constrained architectural components (center-surround antagonism, local receptive fields, tuned normalization, and cortical magnification). Composite Model B contained all architectural components, with the exception of center-surround antagonism. For baseline and single-component models, mean accuracies (± one standard deviation) are reported, where each trial was associated with a distinct base model from the repeated trials of section 4.1.

# I   Code Availability

Code and materials required to reproduce the work presented in this paper are available at `github.com/bionicvisionlab/2023-Pogoncheff-Explaining-V1-Properties`.

