# OpenReview forum: "Explaining V1 Properties with a Biologically Constrained Deep Learning Architecture"
_NeurIPS.cc/2023/Conference — NeurIPS 2023 poster_

### Official Review · Reviewer_Y5Aa · 2023-07-05

**Soundness:** 3 good
**Presentation:** 3 good
**Contribution:** 3 good
**Rating:** 6
**Confidence:** 4

**Summary:**

In this study the authors propose the incorporation of mechanistic biologically inspired filtering and normalization components in deep convolutional networks (DCNs) with the goal of increased alignment of model responses to V1 neural responses and tuning properties. The authors add center-surround receptive fields, local receptive fields, tuned divisive inhibition and cortical magnification to DCNs that are trained with downscaled ImageNet-64x64. The authors perform an extensive ablation of the components above to show their relative importance for alignment with V1 neural responses. The proposed best-performing model produces quite significant improvements on Brainscore's V1 model alignment. The authors also touch upon whether the above models with high V1 alignment are more robust to perceptual distortions.

**Strengths:**

+ The authors propose a unique combination of biologically inspired components that have been shown to exist in primate early visual mechanisms. These components are also explained in good depth and clarity for readers who may not be familiar with the specific computations. To the best of my knowledge, even though each of these components by themselves are not novel, the unique combination explored here appears to be novel and not explored before from the perspective of bettering alignment to V1 properties.
+ The proposed model is significantly outperforming the previous SOTA on explaining neural activity and tuning properties from the Brain-score dataset. The authors have run multiple simulations with different random seeds and add more credibility to their observed findings.

**Weaknesses:**

- As the authors have pointed out, there is a significant drop in accuracy with respect to image classification accuracy. I believe the authors must try to address, from their perspective, why this drop occurs. It is common in this area that models that are trained to better represent biological neural activity tend to suffer from poor classification accuracy. Addressing this issue will be quite a strong contribution.
- The scope of this work is quite limited; I am unsure if the produced improvement on alignment to V1 neural responses and tuning properties is sufficient as the only major contribution in this work.
- It may help if the authors could please add some intuition about why each of the explored components help in improving neural predictivity; it is also important to find out whether these components are useful to improve model-brain alignment regardless of the underlying architecture.

**Questions:**

Please see my review in the Weaknesses section for questions and suggestions.

**Limitations:**

Yes, the authors have adequately addressed limitations in their work.

---

> ### Author Rebuttal · Authors · 2023-08-09
>
> We thank the reviewer for their insights, all of which have helped us improve this work. We ran additional experiments to answer questions about the drop in classification accuracy and reveal insights about the contribution of each component to explaining neural activity. In the latter experiment, we studied the features learned by each model by generating images that maximally activated neurons in artificial V1 layers via gradient ascent, and analyzed the learned parameters of the trainable, biologically-inspired components. We address the concerns raised by the reviewer and elaborate on the results of these experiments below.
>
> > "...there is a significant drop in accuracy with respect to image classification accuracy... Addressing this issue will be quite a strong contribution."
>
> Although not a focal point of this work, we agree that preserving image classification accuracy would be valuable to the targeted research audience. The reduction in accuracy primarily resulted from the introduction of cortical magnification, which assumed that the model’s gaze was focused on the center of the image. As a result, visual stimuli at greater distance from the image center are under-sampled during the polar transformation (Fig. 1 of rebuttal PDF). Images in which the object of interest is not located at the image center became more challenging to classify, reducing model accuracy. This challenge could be mitigated with a mechanism that dynamically determines the fixation center of this polar transform (e.g., saliency or attention maps) or performs inference using multiple fixations. We demonstrate the efficacy of this strategy with a naive proof of concept in which classification is performed by averaging predictions from the five static crops of each image. This simple strategy improved the validation accuracy of the network with all components from 55.1% to 59.9% (without affecting V1 scores). A more sophisticated, dynamic strategy (DOI:10.48550/arXiv.1709.01889) could further reduce this accuracy drop. We have added these details to our manuscript.
>
> > "I am unsure if the produced improvement on alignment to V1 neural responses and tuning properties is sufficient as the only major contribution in this work."
>
> While our results did focus on architecture-driven improvements to model-V1 alignment, this work has further contributions that we briefly summarize below and will detail in a camera-ready version:
> - As the reviewer noted, we systematically analyzed the contribution of biologically-inspired components to explaining V1 activity. This analysis revealed complementary interactions of these components that improved model-V1 alignment beyond what would be suggested by any individual component.
> - The developed models (the most accurate models of V1 to date) are in-silico platforms for analyzing processing in V1. Such image-computable models can enable neuroscientists to study complex dynamics of large neural populations that aren’t readily observable through data-limited, time-consuming, in-vivo observations (DOI:10.1126/science.aav9436) or run surrogate experiments that cannot be done with humans. Their processing and learned parameterization suggest new hypotheses about processing in V1 and provide evidence and alternative views for existing theories. As an example, learned features in networks with tuned normalization were commonly more diverse than baseline those of ResNets (detailed further in response #3). The improvements in V1 property scores that we observed from networks with this component can be treated as additional evidence that competition among neurons driven by tuned normalization gives rise to diverse tuning properties.
> - Improving model-V1 alignment does not trivially improve image classification robustness. While small improvements to corruption robustness from tuned normalization layers were observed, alternative components stood as counterexamples to prior works that have suggested strong correlations between model-V1 alignment and classifier robustness to corrupted images.
>
> > Re: intuition about the contribution of each explored component.
>
> Data-driven approaches have elucidated our strongest intuitions about the contributions of these components and have also produced new neuroscientific insights about processing in V1. We briefly summarize findings from these analyses in the points below, all of which will be expanded upon in the camera-ready paper.
> - Center-surround antagonism improves spatial frequency properties by learning features that are selective to a high variance of spatial frequencies. Most trainable DoG kernels learned low-variance center gaussians, suggesting strong preferences for high frequency patterns and textures (Fig. 2 of rebuttal PDF).
> - We theorized that local RFs would improve response selectivity properties of artificial neurons by removing weight sharing. Our ablation studies demonstrated a drop in the response selectivity property score when local receptive fields were omitted. Surprisingly, this was not observed in isolated-component evaluation.
> - Ablation studies support the role of tuned normalization in improving spatial frequency, response selectivity, receptive field size, surround modulation, and response magnitude tuning alignment. Inter-neuron competition resulting from tuned normalization led to a more diverse feature set (qualitatively depicted in Fig. 2 and quantitatively shown by statistically lower perceptual similarities), likely contributing to these property score improvements.
> - Given the retinotopic organization of V1, we hypothesized that cortical magnification would give rise to better-aligned response selectivity and receptive field size tuning distributions, meanwhile improving neural predictivity. In each trial where cortical magnification was removed, these respective scores dropped, supporting this hypothesis.
>
> The generalization of these components to different architectures has been planned for future work.

---

> > ### Comment · Reviewer_Y5Aa · 2023-08-17
> > **Thank you for the author rebuttal**
> >
> > I thank the authors for responding to the reviewers' concerns in the rebuttal. I appreciate the authors effort to successfully reduce the drop in ImageNet accuracy using static crops of the images. I do agree with the reviewers that the proposed work will be valuable to neuroscientists as a model of V1 processing, I am slightly improving my score from my pre-rebuttal evaluation of the paper.

---

### Official Review · Reviewer_yZMG · 2023-07-06

**Soundness:** 3 good
**Presentation:** 3 good
**Contribution:** 3 good
**Rating:** 7
**Confidence:** 4

**Summary:**

The paper considers deep networks as a model of the visual stream, specifically V1. The authors systematically study the impact of various biological additions to deep networks on alignment of deep nets' representations with V1 recordings.

**Strengths:**

The paper considers several features of the early visual stream that can be added to deep networks, and tests the influence of those features on image classification performance and V1 alignment. While some individual features have been considered before, the approach and especially ablation studies here are novel and, in my opinion, interesting to the community.

The final result is interesting too: combining all 4 architectural features alone resulted in the best V1 alignment (0.605 vs. 0.594 of the top1 V1 model www.brain-score.org/model/vision/623). Adding adversarial features improved it to 0.629, which seems very significant -- the median V1 score at www.brain-score.org/ is less than 0.5.

**Weaknesses:**

The results in Tab. 3 suggest that V1-like features significantly hurt ImageNet performance -- the best V1 model is 16% less accurate than the best ImageNet model. This is mostly due to adversarial training. I think the authors should discuss why it has such an effect.

Two important ablation studies are missing:
1. Adversarial training only, since it has a big effect on both V1 alignment and ImageNet performance.
2. Untrained networks with all/some biological features. All discussed features change the distribution of neural responses even in untrained networks, so it might be that V1 improvements come from that distribution change alone, not from training with those features.

I also suggest the authors include, at least in the appendix, Brain-Scores for other areas (V2, V4, IT) and behavioural data. This is the standard way to evaluate models on Brain-Score, so having all results would make comparisons to other models easier.

**Questions:**

Is it possible to provide a baseline for the "best possible" V1 score by comparing Brain-Score neural data to itself (e.g. with K-fold cross-validation)? I don't think it was done in the original Brain-Score papers, so it's definitely not a hard requirement here. But it would be a great addition.

### Minor issues

> [Line 20] Advances in neuroscience have long been proposed as essential to realizing the next generation of artifical intelligence (AI).

Is that true? I'm not quite sure… also misspelled artificial

> [22] (e.g, convolutional neural networks and mechanisms of attention) owe their origins to biological intelligence

Conv nets need a citation; attention too, and I’m not sure if attention mechanism in transformers were even inspired by biology on the implementation level (see https://www.frontiersin.org/articles/10.3389/fncom.2020.00029/full which says "While the spirit of
attention in machine learning is certainly inspired by psychology,
its implementations do not always track with what is known
about biological attention, as will be noted below.")

Overall: \cite doesn’t generate links to bibliography?

> [45] In specific

I don’t think this phrase is commonly used. “Specifically” or “in particular” would read better.

> [151] these DoG convolutions were only applied to a fraction of the input feature map

Why?

Fig. 1D can benefit from a more detailed explanation. I want to say the original image is on the right but then the transformation doesn’t preserve retinotopy.

> [209] alternate

Alternative

Color-coding Tab. 3 would be great!

The code is in the supplementary, so the authors should indicate it in the main text (and perhaps add it to github afterwards)

### Rebuttal acknowledgement
I have read the rebuttal and responded to the authors. I think it addressed all (minor) concerns that I had, and I still think 7 (accept) is an appropriate score.

**Limitations:**

The limitations and impacts are addressed.

---

> ### Author Rebuttal · Authors · 2023-08-09
>
> We thank the reviewer for their insightful feedback and questions. Our responses to the concerns raised are provided point-by-point below.
>
> >The results in Tab. 3 suggest that V1-like features significantly hurt ImageNet performance -- the best V1 model is 16% less accurate than the best ImageNet model. This is mostly due to adversarial training. I think the authors should discuss why it has such an effect.
>
> While the focus of our work was to highlight the contribution of a biologically-inspired architecture towards explaining V1 activity in response to arbitrary image stimuli, we also realize that preserving image classification accuracy would be valuable to practitioners who wish to use such models in experiments for which core object recognition inference is important. There are two primary reasons for the reduced accuracies:
> 1. In comparison to the 64x64 ResNet50 baseline, the observed reduction in image classification accuracy of models that were not adversarially trained primarily resulted from the introduction of cortical magnification, which assumed that the model’s gaze was focused on the center of the image; consequently, visual stimuli at greater distance from the image center are under-sampled during the polar transformation (Fig. 1 of rebuttal PDF). Images in which the object of interest is not located at the image center (common in ImageNet) became more challenging to classify, reducing model accuracy. This challenge can be mitigated with a mechanism that dynamically determines the fixation center of this polar transform (e.g., saliency or attention maps) or performs inference using multiple fixations. We demonstrate the efficacy of this strategy with a naive proof of concept in which classification is performed by averaging predictions from the five static crops of each image. This simple strategy improved the validation accuracy of the network with all components from 55.1% to 59.9% (without affecting V1 scores). A more sophisticated, dynamic strategy could further reduce this accuracy drop.
> 2. The accuracy of the model that achieved the highest model-V1 alignment (the adversarially trained, biologically constrained model) was further reduced by adversarial training.  We evaluated the classification accuracy of an adversarially trained ResNet50 at 55.5% (rebuttal PDF, Table 2).
>
> These revisions will be detailed and clarified in the camera-ready paper.
>
> >Two important ablation studies are missing: 1) Adversarial training only and 2) Untrained networks with all/some biological features
>
> We thank the reviewer for these suggestions and agree that they are important. We have run these evaluations and included the results in Tables 1 and 2 of the rebuttal PDF. For ease of reference, the V1 Overall Brain-Score for the adversarially trained ResNet50 was 0.581 and no statistically significant correlation was observed between untrained and trained model V1 Overall scores. These evaluations will be added to the paper.
>
> >I also suggest the authors include, at least in the appendix, Brain-Scores for other areas (V2, V4, IT) and behavioral data. This is the standard way to evaluate models on Brain-Score, so having all results would make comparisons to other models easier.
>
> We agree with the reviewer’s suggestion. These scores will be added to our supplementary material. For reference, V2, V4, and IT scores for the top performing model (all-components with adversarial training) are .343 (rank 26), .459 (rank 118), and .343 (rank 168), respectively.
>
> > Question: Is it possible to provide a baseline for the "best possible" V1 score by comparing Brain-Score neural data to itself (e.g. with K-fold cross-validation)? I don't think it was done in the original Brain-Score papers, so it's definitely not a hard requirement here. But it would be a great addition.
>
> V1 Predictivity scores as computed as correlations between measured and predicted neural activity, normalized by internal consistency of the measured neural data.  V1 Property scores are similarly ceiled according to maximum distribution similarities observed among the measured neural data.\
> Regarding these internal consistency scores and maximum distribution similarities, we unfortunately could not calculate this as we do not have access to the private neural evaluation data. This is an interesting question, however, and the answer to it could have interesting implications regarding what we could expect from an “optimal” model.
>
> > Minor: [151] these DoG convolutions were only applied to a fraction of the input feature map. Why?
>
> We skip the DoG convolution for some channels to account for the fact that not all V1 neurons have a symmetric surround suppression region (DOI:10.1523/JNEUROSCI.19-23-10536.1999, 10.1038/nn1310).
>
> >Minor: Fig. 1D can benefit from a more detailed explanation. I want to say the original image is on the right but then the transformation doesn’t preserve retinotopy.
>
> We apologize for this confusion. The original image is on the left and the transformed image is on the right. Fig. 1 of the rebuttal PDF shows two example images before and after the transformation. The caption of Figure 1D has been updated and these example figures will be added to the supplementary material for clarity.
>
> >Minor: The code is in the supplementary, so the authors should indicate it in the main text (and perhaps add it to github afterwards)
>
> We plan to make our code publicly available on github following anonymous review and will add the corresponding link in the main text.
>
> >Minor: Remaining minor issues regarding grammar, spelling, formatting, citation issues, and unclear claims about the importance of neuroscience in AI and biologically inspired mechanisms in artificial neural networks.
>
> We would like to thank the reviewer for highlighting these oversights. We will resolve these issues and clarify these claims in the camera-ready paper.

---

> > ### Comment · Reviewer_yZMG · 2023-08-10
> > **Response to rebuttal**
> >
> > Thank you for the response! Overall, I think it addressed all (minor) concerns that I had, and I still think 7 (accept) is an appropriate score.
> >
> > > We demonstrate the efficacy of this strategy with a naive proof of concept in which classification is performed by averaging predictions from the five static crops of each image
> >
> > Great! I believe this technique, test-time augmentation, is not uncommon in deep learning, and makes sense for models of the visual stream.
> >
> > I also think the new results with untrained networks, which achieved much lower V1 alignment than the trained ones, strengthen the results -- they suggest that all added components don’t just match superficial features of V1 processing, but lead to better (in terms of alignment to V1) training.
> >
> > Other reviewers have noted the limited scope of this paper, but I don’t completely agree. I think the contribution of this paper is significant enough for the task of building better models of the visual stream.

---

### Official Review · Reviewer_EW9X · 2023-07-06

**Soundness:** 3 good
**Presentation:** 3 good
**Contribution:** 3 good
**Rating:** 5
**Confidence:** 4

**Summary:**

The authors incorporated four well-known architectural components of V1 into an earlier layer of the CNN, resulting in a reduction in task performance but an improved alignment with V1 neurons' behaviors. Their study demonstrated that cortical magnification led to the most significant enhancement in alignment, as observed in the overall property and predictability of V1 in the Brain-Score benchmark test. Tuned normalization also improved alignment in certain V1 properties, while the contribution of center-surround mechanisms appeared to be minimal or data-dependent. These improvements generally ranged between 1% and 2%.

**Strengths:**

The motivation and hypotheses of the study are reasonable, and the exploration is conducted in a systematic and logical manner. The finding that cortical magnification provides some improvement in alignment is interesting.

**Weaknesses:**

The introduction of brain architectural components was expected to enhance the alignment of the model with V1 data in the Brain-Score test, so the results are not particularly surprising. While the paper contains some systematic and well-done experiments, it does not provide an explanation as to why certain architectural components would have specific effects. As a result, the main contribution is simply showing  incorporating more information relevant to the data would improve performance in explaining the data, at the expense of the model's task performance. Although the work may hold value, its contribution might not reach the level typically expected in a NeurIPS paper.

**Questions:**

It would be helpful to discuss why and how certain architectural components would  produce greater alignment while others do not.  Is there any theoretical and conceptual framework that can help us to make sense of the results?

**Limitations:**

Limitations on performance drop has been discussed.  The work could have implications on neuroscience.

---

> ### Author Rebuttal · Authors · 2023-08-09
>
> We thank the reviewer for their insights. We have broken down the raised concerns and provide our responses below.
>
> >The introduction of brain architectural components was expected to enhance the alignment of the model with V1 data in the Brain-Score test, so the results are not particularly surprising
>
> We appreciate this feedback but suggest the contrary, that the results are nontrivial and surprising. From one perspective, classical neuroscientific models of V1 (inherently based on architectural components of the brain) fail to predict V1 neural responses as well as task-driven ANNs (DOI:10.1101/2021.03.01.433495). Second, none of the top-scoring models on Brain-Score feature biological components and prior work has demonstrated strong correlations between ImageNet accuracy and model-brain alignment (DOI:10.1101/407007). Further, in isolation (main text, Table 1), most modules did not improve model-V1 alignment (center-surround antagonism and both divisive normalization components reduced the V1 Overall score, on average). Surprisingly, it was only when these components were combined that we observed drastically improved explanations of V1, suggesting their complimentary contribution.
>
> >it does not provide an explanation as to why certain architectural components would have specific effects
>
> We thank the reviewer for this insight and have since run additional studies to explain these observations. Specifically, we analyzed the features learned by each network variant by visualizing images that maximally activated neurons in artificial V1 layers via gradient ascent and studied the learned parameters of trainable, biologically-inspired components. We summarize the conclusions of these experiments below and will further detail these insights in the camera-ready paper:
> - Center-surround antagonism improves spatial frequency properties by learning features that are selective to a highly varying spatial frequencies. Most trainable DoG kernels learned low-variance center gaussians, suggesting strong preferences for high frequency patterns and textures (Fig 2 of rebuttal PDF).
> - We theorized that local RFs would improve response selectivity properties of artificial neurons by removing weight sharing. Our ablation studies demonstrated a drop in the response selectivity property score when local receptive fields were omitted. Surprisingly, this was not observed in isolated-component evaluation.
> - Single and multi-component studies support the role of tuned normalization in improving spatial frequency, response selectivity, receptive field size, surround modulation, and response magnitude tuning alignment. Inter-neuron competition resulting from tuned normalization led to a more diverse feature set (qualitatively depicted in Fig 2 and quantitatively shown by statistically lower perceptual similarities), likely contributing to these property score improvements.
> - Given the retinotopic organization of V1, we hypothesized that cortical magnification would give rise to better-aligned response selectivity and receptive field size tuning distributions, meanwhile improving neural predictivity. In each trial where cortical magnification was removed, these respective scores dropped, supporting this hypothesis.
>
> >the main contribution is simply showing incorporating more information relevant to the data would improve performance in explaining the data, at the expense of the model's task performance
>
> While our results did focus on architecture-driven improvements to model-V1 alignment, it has further contributions that we briefly summarize below and will expand upon in the camera-ready paper:
> - As the reviewer noted, we systematically analyzed the contribution of biologically-inspired components to explaining V1 activity.
> - The developed models (the most accurate models of V1 to date) are in-silico platforms for analyzing processing in V1. Such image-computable models can enable neuroscientists to study complex dynamics of large neural populations that aren’t readily observable through data-limited, time-consuming, in-vivo observations (DOI:10.1126/science.aav9436) or support experiments that cannot be run in humans. Their processing and learned parameterization suggest new hypotheses about processing in V1 and provide evidence and alternative views for existing theories. As an example, learned features in networks with tuned normalization were commonly more diverse than baseline those of ResNets. The improvements in V1 property scores that we observed from networks with this component can be treated as additional evidence that competition among neurons driven by tuned normalization gives rise to diverse tuning properties.
> - Improving model-V1 alignment does not trivially improve image classification robustness. While small improvements to corruption robustness from tuned normalization layers were observed, alternative components stood as counterexamples to prior works that have suggested strong correlations between model-V1 alignment and classifier robustness to corrupted images.
>
> Regarding diminished classification accuracy, these reductions primarily resulted from the introduction of cortical magnification, which assumed that the model’s gaze was focused on the center of the image. Images for which the object of interest is not located at the image center became more challenging to classify, reducing accuracy (rebuttal PDF, Fig 1). This challenge can be mitigated with a mechanism that dynamically determines the fixation center of this polar transform (e.g., saliency or attention maps) or performs inference using multiple fixations. We demonstrate the efficacy of this approach with a naive strategy that performs inference by averaging predictions from five static crops of each image, improving the accuracy of the network with all components from 55.1% to 59.9% (without affecting V1 scores). A more sophisticated, dynamic strategy (DOI:10.48550/arXiv.1709.01889) could further reduce this accuracy gap.

---

> > ### Comment · Reviewer_EW9X · 2023-08-20
> > **Thanks for the responses**
> >
> > Thank you for the responses. It is indeed interesting to show that "it was only when these components were combined that we observed drastically improved explanations of V1, suggesting their complimentary contribution."  Somehow I did not get this on my earlier reading.
> > Thank you also for the additional experiments which do provide some insights.
> > I am willing to upgrade my score, but my overall sentiment is very much aligned with that of Reviewer  bbGj.
> >
> > Incidentally, it would also be interesting to compare this transfer-learning + neural constraint model with purely data-driven model, e.g.
> > https://journals.plos.org/ploscompbiol/article?id=10.1371/journal.pcbi.1006897
> > However, I suppose that data set won't have contextual modulation effects, and thus can't serve as a real neuron-in-silico.
> > Perhaps some Allen Institute dataset on mice could work.

---

### Official Review · Reviewer_bbGj · 2023-07-07

**Soundness:** 2 fair
**Presentation:** 2 fair
**Contribution:** 3 good
**Rating:** 5
**Confidence:** 4

**Summary:**

This paper incorporates a wide range of biologically inspired components into the initial stages of a ResNet model to see if these result in improved alignment with properties of V1 neurons.  Specifically, the authors incorporate architectural components for Center-Surround, Local Receptive Fields, Divisive Normalization, Tuned Normalization, and Cortical Magnification. They find that some of these components are complementary when explaining different properties of V1. Adversarial training further improves the match to V1 responses.

**Strengths:**

This paper provides a through discussion about many hypothesized computations in V1, and incorporates these computations into deep neural networks as modules with learnable components. The individual contributions of these components are systematically evaluated in terms of how well they capture properties of V1 and predict V1 responses to stimuli. Building this type of biologically inspired model with many V1 components is novel and the discussion of incorporating biological components into AI systems is currently of high interest for the NeurIPS audience (both on the CS and neuroscience communities).

**Weaknesses:**

* The idea of including biological components into neural networks is not novel, and the paper lacks full discussion of other attempts along these lines. Perhaps most relevant here is VOneNet (Dapello et al. 2020) which incorporates properties of V1 into a convolutional neural network (this paper is cited but not in the context of modeling V1). Although the exact architectural components added are different here (and designed for specific response properties), the paper would benefit from an explicit discussion about what sets this work apart from what was previously done.
* Re clarity: As the paper is currently written, I found difficult to follow what is previous work and what is built into the model. In some parts of the “Background” it is mentioned that certain things are adopted in the previous work, but other parts are not discussed as being incorporated until the Methods. It might help the reader if these sections were restructured so that it is clear how the previous work builds into the proposed architecture.
* The paper claims to “introduce architectural components based on neuroscience foundations” however, from what I can tell, all of the included architecture components have been previously proposed. The novelty of this paper is including all of them within one model and analyzing them systematically.
* Due to the number of models and response properties that are tested, and the somewhat small differences in many cases, it is difficult to interpret which of the brain-score results are significant, although I do appreciate that the authors report mean and standard deviation across 3 trials of training and evaluation. Perhaps just changing the colors to make it clear which differences are or are not significant (correcting for multiple comparisons) would help? But generally, the changes in Brain-Score seems relatively small as all models are still far from the noise ceiling, making some of the claims of the paper seem too grandiose given the reported results.
* The robustness to corrupted images experiment also seems like the differences from the baseline model may be within the noise after correcting for multiple comparisons, and the overall changes are very small.


**Questions:**

a)	It would be helpful to include a model with *just* adversarial training and none of the biological components as an additional baseline model in Table 3.

b)	I’m a bit confused about the greedy backwards elimination approach. Was the architectural component that “reduced overall V1 alignment” selected specifically at that given stage of the architecture (ie was a model trained eliminating each component separately, and then just one was chosen?) Or was this elimination order based on the single-architectural component experiment?

c)	On line 199-200 it states that the cortical magnification is applied immediately before the ResNet50 layer 1. Does this ResNet still have the first convolutional layer that proceeds all of the residual blocks, such that the cortical magnification is acting on the output of the convolutional layer/pooling? This detail should be clarified.

d)	Was the decision to downsample the ImageNet images to 64x64 based on biology or computational constraints? Is it possible to train on more standard image sizes? This would possibly help the accuracy of the models.

e)	Related to the above – is the baseline ResNet50 presented trained on these 64x64 images as well?

f)	What networks are being referred to on lines 327-330? Are these the networks with adversarial training? (If so, this claim maybe should be toned down, as the margins are not “large” and still far off from the noise ceiling).

Minor:
* Lines 25-28: It would be helpful if specific “neuroscientific” models were listed here. I think the idea is that the models referred to in this section are hand-designed based on observed neural properties, rather than being optimized in some other manner?
* Line 32: “Through typical task-driven training alone” – this is not quite true, as a (typically linear) readout must be trained to map the activations of the neural network onto the responses of neurons.
* Lines 36: It would improve the paper if citations showing that CNNs are not achieving properties of the visual system were included here.
* Lines 68-73 are vague and uninformative to the context of the presented work. It would help to be more specific about failures of the models and previous work.
* Line 325: Include a reference to the supplementary section documenting the training details for the adversarially trained network.


**Limitations:**

The authors discuss the limitations in the discussion.

---

> ### Author Rebuttal · Authors · 2023-08-09
>
> We thank the reviewer for their insights. Our responses to each raised concern are provided in the points below and we will add these details to the camera-ready paper.
>
> >Re: The paper would benefit from an explicit discussion about novelty
>
> We appreciate this feedback and have run additional experiments to expand our results and discussions. We studied the features learned by each network by generating maximally activating images of artificial V1 layers via gradient ascent, analyzed learned parameters of trainable components, and elaborated on the original ablation studies. We summarize our findings in the points below.
> - Center-surround antagonism improves spatial frequency properties via features selective to a high variety of spatial frequencies. Most trainable DoG kernels learned low-variance center gaussians, suggesting strong preferences for high frequency patterns and textures (rebuttal PDF, Fig 2).
> - We theorized that local RFs would improve response selectivity of artificial neurons by removing weight sharing. Ablation studies demonstrated a drop in the response selectivity property score when local receptive fields were omitted. Surprisingly, this was not observed in isolated-component evaluation.
> - Ablation studies supported the role of tuned normalization in improving spatial frequency, response selectivity, receptive field size, surround modulation, and response magnitude tuning alignment. Inter-neuron competition resulting from tuned normalization led to a more diverse feature set (qualitatively depicted in Fig 2 and quantitatively supported by statistically lower perceptual similarities), likely contributing to these improvements.
> - Given the retinotopic organization of V1, we hypothesized that cortical magnification would give rise to better-aligned response selectivity and receptive field size tuning distributions and additionally improve explanation of neural response. Each of these scores dropped whenever cortical magnification was removed, supporting this hypothesis.
>
> Regarding explicitly identifying the novelty of this work:
> - Prior works have evaluated a subset of these components independently. Our systematic analysis revealed nontrivial and complementary interactions that improved model-V1 alignment beyond what would be suggested by individual components, yielding SOTA models of macaque V1.
> - The developed models are in-silico platforms for analyzing processing in V1. Such image-computable models can enable neuroscientists to study complex dynamics of large neural populations that aren’t readily observable through data-limited, time-consuming, in-vivo observations (DOI:10.1126/science.aav9436) or support experiments that cannot be run in humans. Their processing and learned parameterization suggest new hypotheses about processing in V1 and provide evidence and alternative views for existing theories.
> - Improving model-V1 alignment does not trivially improve classifier robustness. While small improvements to corruption robustness from tuned normalization layers were observed, alternate components stood as counterexamples to trends suggested in prior works.
>
> >Re: Interpretability of brain-scores across many models and small changes in many cases
>
> We wish to clarify that integrating individual components had little impact on model-V1 alignment (with the exception of cortical magnification). It was when these components were integrated together that substantial improvements emerged (rebuttal PDF, Fig 3). To date, these are the most accurate models of macaque V1 and prior work has demonstrated that even models with neural alignment far from the noise ceiling have utility in revealing novel insights about processing in the brain (DOI:10.1126/science.aav9436).
>
> >Re: Small changes in robustness to corrupted images
>
> We agree that the changes resulting from tuned normalization were minor (akin to the architecture-only effects observed in VOneNet (Dapello et al. 2020)). Notably, different components challenged the presumed link between model-V1 alignment and classifier robustness. Systematically investigating this divergence through analysis of the training dataset, training dynamics, and architecture would be an intriguing avenue for future exploration.
>
> >Re: Adversarially trained ResNet50 as a baseline
>
> We appreciate this suggestion. These results have been included in Table 2 of the rebuttal PDF.
>
> >Re: Greedy backwards elimination
>
> In this approach, we iteratively removed individual components from the architecture, computed which contributed the least to the V1 Overall score, and then removed this single component. This was done in a top-down approach, starting with our top-performing model, to deduce the critical components without having to evaluate every model permutation.
>
> >Re: Cortical magnification integration
>
> This ResNet still has the first conv and batch norm layers.  The cortical magnification layer replaced the pooling layer before the first residual block, as it implicitly performs pooling among pixels of the same polar cell.
>
> >Re: ImageNet downsampling
>
> This downsampling was done for all models due to computational constraints (not a requirement of any component). We agree that training without downsampling is likely to improve model accuracy. Reductions in model accuracy were also found to be mitigated by multi-fixation inference strategies that addressed classification challenges associated with cortical magnification.
>
> >Re: Toning down claim on lines 327-330
>
> The networks in question refer to the top performing models with and without adversarial training. Top V1 alignment scores were previously separated by small margins, and these models were evaluated as the most accurate models of activity in V1 to date. These substantial improvements are depicted in Fig 3 of the rebuttal PDF.
>
> >Re: Paper clarity and minor suggestions
>
> We appreciate these suggestions, agree with the reviewer’s points, and will address these points in the camera-ready paper.

---

> > ### Comment · Reviewer_bbGj · 2023-08-14
> > **Response to author rebuttal**
> >
> > Thank you for the response. From this, it is more clear that the benefit of this work comes from integrating all of the different V1 components into the same model, showing that this results in better alignment with V1 responses. If this is the case, I think that should be made more explicit throughout the paper (ie in the listed bullet points on line 46).
> >
> > I also agree with Reviewer yZMG about the Untrained models being a nice addition.
> >
> > After the author responses I still find this work borderline. I appreciate that amount of effort it takes to integrate all of these components into a single model (and that itself may be useful for the field as a new baseline, as the authors discuss). But as currently presented, I'm not sure if that is "new" enough or if the insights gained are deep enough for a typical NeurIPS paper.
> >
> > Finally, this is a bit of my personal preference in terms of wording (and thus is not influencing my score), but as neuroscientists, I wonder if we want to fall into the trap of publishing paper after paper highlighting the achievements of "SOTA" on a particular benchmark when the improvements are around +2-3%? To me, it seems especially problematic given the limited size of the current datasets. This is why I was highlighting the concerns about multiple-comparisons above, and suggestions to tone down the language about things like "unprecedented explanation" of V1.

---

### Author Rebuttal · Authors · 2023-08-09

We thank the reviewers for their insightful and productive feedback. We were grateful to read that the reviewers agreed that the biologically-constrained models proposed in this work significantly outperform previous SOTA on explaining neural activity observed in macaque V1 (Y5Aa, yZMG), were systematically evaluated (EW9X, bbGj, yZMG), and that the work is of of high interest to the NeurIPS audience (bbGj, Y5Aa).

The reviewers raised constructive questions regarding intuition behind the contribution of each analyzed component towards explaining neural activity in V1 (Y5Aa, bbGj, EW9X), solutions to mitigate reductions in image classification performance upon introducing these biologically-motivated components (Y5Aa, yZMG, EW9X), and further implications of the observed results (Y5Aa, bbGj, EW9X).

We address the questions and concerns raised by each reviewer point-by-point in the respective threads below. In summary, data-driven insights emergent from studying learned features and parameters of the trained models are suggestive of each component's contribution to explaining neural activity in V1. Static center-of-gaze assumptions of the cortical magnification layer made the model more susceptible to misclassifying images for which the object of interest was outside of the image center, a challenge that could be mitigated with dynamic or recurrent fixation strategies. Finally, the learned parameters and processing strategies of these state-of-the-art models of V1 could unveil hypotheses and provide evidence for visual processing strategies in the primary visual cortex.

---

### Decision · Program_Chairs · 2023-09-21

**Decision:**

Accept (poster)

**Comment:**

This paper explores the addition of biologically plausible architectural components to CNNs.
Ablation analyses examine the impact of the different components.    The model achieves top
scores on the Brain-Score V1 benchmark.